# Enteric Chromosomal Islands: DNA Packaging Specificity and Role of λ-like Helper Phage Terminase

**DOI:** 10.3390/v14040818

**Published:** 2022-04-15

**Authors:** Helios Murialdo, Michael Feiss

**Affiliations:** 1Fundación Ciencia & Vida, Av. Zañartu, Santiago 1482, Chile; 2Department of Microbiology and Immunology, Carver College of Medicine, University of Iowa, Iowa City, IA 52242, USA

**Keywords:** PICI hijacking, phage packaging of chromosomal islands, terminase interactions, virus DNA packaging

## Abstract

The phage-inducible chromosomal islands (PICIs) of Gram-negative bacteria are analogous to defective prophages that have lost the ability to propagate without the aid of a helper phage. PICIs have acquired genes that alter the genetic repertoire of the bacterial host, including supplying virulence factors. Recent work by the Penadés laboratory elucidates how a helper phage infection or prophage induction induces the island to excise from the bacterial chromosome, replicate, and become packaged into functional virions. PICIs lack a complete set of morphogenetic genes needed to construct mature virus particles. Rather, PICIs hijack virion assembly functions from an induced prophage acting as a helper phage. The hijacking strategy includes preventing the helper phage from packaging its own DNA while enabling PICI DNA packaging. In the case of recently described Gram-negative PICIs, the PICI changes the specificity of DNA packaging. This is achieved by an island-encoded protein (Rpp) that binds to the phage protein (TerS), which normally selects phage DNA for packaging from a DNA pool that includes the helper phage and host DNAs. The Rpp–TerS interaction prevents phage DNA packaging while sponsoring PICI DNA packaging. Our communication reviews published data about the hijacking mechanism and its implications for phage DNA packaging. We propose that the Rpp–TerS complex binds to a site in the island DNA that is positioned analogous to that of the phage DNA but has a completely different sequence. The critical role of TerS in the Rpp–TerS complex is to escort TerL to the PICI *cosN*, ensuring appropriate DNA cutting and packaging.

## 1. Introduction

The λ-like phages are a family of tailed, dsDNA bacteriophages. During intracellular proliferation, the phages of this family produce empty icosahedral capsids (proheads or procapsids) and tails. Phage DNA replication generates concatemers: end-to-end multimers of the unit-length virion DNAs. Processing concatemeric DNA into unit-length virion genomes occurs during packaging. Processing involves recognition of a specific sequence, called *cos*, by a phage-encoded packaging enzyme, terminase. Terminase cuts concatemeric DNA at a *cos* and translocates the DNA into the capsid. Terminases, in general, are hetero-multimers of a large subunit, TerL, and a small subunit, TerS. TerL contains the endonuclease and translocase activities, and TerS directs TerL to *cos*, positioning TerL’s endonuclease to accurately cut *cos*.

Recent publications from the Penadés lab show that Gram-negative bacteria carry a class of mobile genetic elements called PICIs for phage-induced chromosomal islands [1,2]. In a cell with an actively multiplying λ-like phage, a “helper phage”, PICI elements induce, excise from the bacterial chromosome, and replicate. Subsequently, PICIs hijack the DNA packaging system of the helper phage, enabling packaging of the PICI DNA into functional virions that can transfer the PICI to a new bacterial host. The Penadés group has studied four PICIs, namely A, B, C, and G [1,2] (Appendix A), and worked out the molecular basis of hijacking. These findings have implications for understanding λ-like phage DNA packaging. Here, we sketch out the phage DNA packaging process and briefly review the procedure by which PICIs snatches the λ-like DNA packaging machinery and discuss hijacking from a phage-oriented perspective.

### 1.1. DNA Packaging in the λ-like Phages

Terminase-*cos* interactions orchestrate a complex series of steps required for DNA packaging. Phage λ’s *cos* (*cos*-λ) has three elements: *cosQ*, *cosN,* and *cosB* (Figure 1A–C). *cosN* and *cosB* are required to initiate DNA packaging [3]. *cosB* has three closely related sequences, namely R3, R2, and R1, to which TerS binds to initiate DNA packaging. Between R3 and R2 is I1, a binding site for *E. coli*’s sequence-specific DNA bending protein, IHF. TerL, anchored by *cosB*-bound TerS, nicks the two DNA strands of *cosN* at positions staggered 12 base pairs apart (Figure 1A). Nicking generates the 12-base-long, complementary cohesive ends of mature viral DNAs. The newly generated cohesive ends are separated by terminase. Terminase remains bound to the *cosB*-containing DNA end. This protein-DNA complex captures a prohead and TerL uses ATP hydrolysis to power translocation of the DNA into the prohead. Once the translocation machinery encounters a second, downstream *cos*, TerL nicks *cosN*, finishing DNA processing and terminating packaging. Termination requires *cosN* and *cosQ*. The prohead shell is an icosahedron with a unique vertex called the portal vertex, which is a dodecamer of radially disposed subunits with a central channel for DNA entry and exit from the prohead [4,5]. TerL, using a C-terminal, portal-binding domain, docks on the portal vertex to start DNA translocation [6].

### 1.2. Terminase Structure

TerS-λ (181 amino acids) has three domains: (1) an N-terminal, winged helix-turn-helix (HTH) DNA-binding domain (DBD) that also forms a stable dimer; (2) a predominately α-helical central oligomerization domain; and (3) a C-terminal functional domain for TerL binding [9]. A λ-21 hybrid phage with a chimeric TerS indicated that the TerL-binding domain is in the C-terminal half of TerS, as follows: Phage 21′s TerS specifically binds *cosB*-21 and TerL-21. The N-terminal half of the chimeric TerS contains the DBD of TerS-21, and the C-terminal half derives from λ and contains the TerL-binding specificity domain [10]. In vitro studies indicate the C-terminal 40 residues of TerS-λ are protease-sensitive [11,12]. Finally, TerS-λ, missing the last 30 amino acids, binds TerL poorly [11]. We note that the TerS arrangement of an N-terminal DNA-binding domain and a C-terminal TerL-binding domain may be an ancient construct. That is, in the P22-related phages, TerS chimeras support a similar domain structure [13], and deleting the last 22 residues of TerS-P22 abrogates the ability to bind TerL-P22 [14].

The large subunit of λ terminase, TerL-λ, (641 amino acids), contains (a) an N-terminal domain (NTD) consisting of TerS-binding domain and a large motor domain that uses ATP to translocate DNA into the prohead and (b) a C-terminal domain (CTD) comprising an endonuclease center followed by an α-helix predicted to form a parallel coiled-coil and at the C-terminus, a portal binding tether (Figure 1B) [6,15,16].

TerS monomers form higher-order multimers that have not been studied in detail [17]. λ terminase, at physiological concentrations, forms a TerS_2_:TerL_1_ heterotrimer called a protomer [6,8]. Protomers, at high concentrations can be assembled into an active, higher-order multimer [6], recently identified as a pentamer of protomers (C. E. Catalano, pers. comm.). This pentamer of protomers is likely to be related to the portal-docked, pentameric translocation motor. In vitro, the pentamer carries out *cosN* cleavage upon the addition of Mg^++^ in the absence or presence of proheads, that is, prior to or after prohead portal docking. Whether or not the initiation of packaging is carried out by a multimer of protomers or a simpler assemblage is not a critical issue for the present discussion.

### 1.3. Terminase:cos-λ Interactions

*cosN* includes the 12 bp corresponding to the cohesive ends of the virion DNA and 2 bp flanking the nick sites. Of these 16 bp, 10 bp display 2-fold rotational symmetry (Figure 1A). The *cosN* symmetry is found in many λ-like phages [18]. This structure suggests that a simple arrangement of terminase subunits may sponsor the initial *cos* cleavage event. Its symmetry indicates that two terminase protomers, symmetrically disposed on *cosN*, likely carry out the nicking reaction (Figure 1B) [10,11]. The protomers are proposed to dimerize through formation of a coiled-coil interface using the α helixes distal to the endonuclease centers (Figure 1B). TerL endonuclease centers are not highly DNA sequence specific, as deletion of *cosB* results in greatly reduced and inaccurate nicking [12]. Thus, TerS-*cosB* interactions are critical for anchoring TerL for accurate and efficient *cosN* nicking. These interactions are analyzed next.

As in the case of *cosB*-λ, the *cosB*s of other λ-like phages, such as φ80, 21, and Gifsy-1, have three R sequences and an IHF site. The order of subsites is R3-I1-R2-R1, where R2 is oriented opposite to R3 and R1. An exception is phage N15, the *cosB* of which consists of a single R sequence, R3-N15, analogous in location to R3-λ [19,20] (Figure 1C and Figure 2).

In the TerS DBD dimer, the α2 recognition helixes, i.e., the DNA-contacting α-helixes, are solvent-exposed and positioned to contact R sequence bps in the major groove. In addition, the wings are positioned to make minor groove contacts [21]. The binding of IHF between R3 and R2 (Figure 1) bends the DNA 180°, positioning the R sequences major grooves facing each other. A compelling model positions a dimer of TerS DBDs docked on R3 and R2 across the IHF-induced bend [22].

Thus, the TerS-*cosB* interaction is reckoned to be a typical interaction between a HTH [23] motif and the major groove of the binding site. The extent of the λ R sequences is partially defined. R2 and R1 of *cosB*-λ share a 16 bp identical sequence, whereas R3 differs from the former by 3 bps. R3 extends from bp 50 to 65, where bp 1 is the first base of the left cohesive end of the virion DNA. NMR studies indicate that R3 bp 55–63 interact with TerS DBD. Genetics studies indicate that: (1) R3 seems to be critical, as a transition mutation in a conserved bp was found to have the most severe phenotype when placed in R3 rather than in R2 or R1 [24], and (2) spacing mutations showed that the *cosN*-to-*cosB* distance is crucial—even deleting 3 bp between *cosN* and *cosB* caused a severe packaging defect [25,26]. Additional bp substitutions have shown that R3 mutations affecting packaging include changes at bp 56, 58, and 59 [24,27]. To summarize, the bp 55–63 segment likely contains the bp specifically contacted by TerS.

The α2 recognition helix of the TerS-λ DBD extends from residue 17–24, and the wing includes residues 31–39. Genetic studies indicate that specific interactions exist between TerS-λ residue E24 and R3 bp 56, which in turn suggests that the α2 helix is oriented with the C-terminal end towards the 5′ end or R3 [27]. An additional TerS–*cosB* interaction is predicted to be a minor groove contact between TerS wing residue Lys-35 and the minor groove likely near but 5-prime of the major groove contacts [21,27,28].

Interestingly, λ and N15 share their packaging specificity. That is, N15 packages λ DNA efficiently [20]. Scanning mutagenesis places N15′s R3 between bp 48 and bp 60. Therefore, bp 55–60 in λ R3 are contacted not only by TerS-λ but also by other λ-like TerS molecules, such as the one encoded by phage N15.

Based on genetic studies indicating the critical nature of R3, it has been assumed that the TerS-λ bound at R3 anchors the TerL that nicks *cosN*’s bottom strand between bps 12 and 13 [18]. It has also been assumed, based on the 2-fold symmetry of some *cosN* bp (see above), that a second TerL sitting 5′ to the TerS-anchored TerL molecule would be responsible for the top strand nicking, generating the protruding 5′ end of the λ chromosome [18].

### 1.4. PICI Packaging by λ-like Helper Phages

The Peñades laboratory has shown that when a PICI-carrying bacterium is undergoing an active lytic development by a λ-like phage, either the result of prophage induction or external infection, the PICI element is induced to excise from the bacterial chromosome and begin DNA replication. The concatemeric PICI DNA is packaged into virions using the virus assembly proteins provided by the helper phage [1,2]. Concomitantly, the helper phage is inhibited from packaging its own DNA. To identify the inhibitory activity, Fillol-Salom et al. [2] screened genes in *E. coli* CFT073, the PICI-A-containing bacterium, for a gene whose product was able to block plaque formation by known helper phages λ or φ80. The RppA-encoding gene, *rppA*, and a nearby functional *cosN* were identified [1]. The layout of the PICI-A *cos*-*rpp* organization is much like the *cos-terS* arrangement in helper phages (Figure 1C), as follows. The PICI-A element’s *cos* contains *cosQ* and *cosN*, with sequences nearly identical to those of the λ-like helper phages (Figure 2). No match to a λ-like phage *cosB* is apparent. That is, (1) no repeated R sequences are found, and (2) no sequence matching the R3 sequence of a helper phage is apparent [1]. An additional three islands, PICI-B, PICI-C, and PICI-G, were found to have the same *cos*-*rpp* organization (Appendix A). Subsequently, RppA was shown to bind to TerS, forming a heterodimer and preventing TerS from binding to the helper phage’s *cosB*. Instead, the Rpp–TerS heterodimer binds to *cos*-PICI for assembly of an initiation complex for DNA cutting and packaging.

Rpps vary in sequence and in length from 139 to 154 amino acid residues (Figure 3). The Rpp secondary structure has six α-helixes and two β-strands (Figure 3). The two N-proximal α-helices and a pair of β-strands form a winged HTH DNA-binding domain similar to λ TerS but with a different DNA sequence specificity. In both Rpp and TerS, a long helix, α3, tethers the DBD to the oligomerization domain [1]. Structural information about TerS and RppC homodimers and the TerS-RppC heterodimer suggests a hierarchy of dimerization affinities, as follows. The TerS DBDs form a tight dimer through interactions between residues 11–13 of α1 with residues 43, 47, and 51 of α3. In contrast, RppC dimerizes through interactions of helixes α3 to α6, especially by an α6–α6 coiled-coil. In the RppC-TerS heterodimer, RppC mimics the DBD interactions of TerS, forming additional coiled-coil interactions with the extended TerS α3 helix and α6 of RppC [1]. The extent of interactions is consistent with the RppC-TerS heterodimer being the most stable of the three dimeric forms although we lack structural information about residues 99 to 181 for TerS. The likely affinities agree with the biological roles of RppC, which blocks TerS from acting and redirects terminase to *cos*-PICI-C. It follows that Rpps are likely made in molar excess of TerS to effectively prevent TerS dimer formation. Interestingly, the positions of the HTH DNA recognition α2 helixes are nearly identical in the RppC-TerS-λ heterodimer and the TerS-λ DBD homodimer [1,22]. The DBD domains of TerS and Rpp proteins are structurally very similar and match that of members of the LysR transcription factors, as pointed out by Fillol-Salom et al. [1]. In contrast, the origins of the oligomerization domains of Rpp proteins may differ from that of the λ-like TerSs (Figure 3).

Here, we look at the published data on the Rpp and TerS interactions in the Rpp–TerS heterodimer with *cos*-PICI and between the Rpp–TerS heterodimer and the helper phage TerL. We make two proposals and a hypothesis.

The first proposal is that there is an R3 equivalent in *cos*-PICI (see Section 2.3). We suggest that a R3-PICI will be in the segment extending from bp 55–63, a segment that is in common in the R3s of phages λ and N15.

The second proposal is that differences in wing structure of the Rpp’s could affect DNA binding specificity to the different PICIs (Section 2.3).

We hypothesize that the role of TerS in the Rpp–TerS heterodimer is to provide a TerL-binding domain (Section 2.4).

## 2. Proposals: Recognition of *cos*-PICI and the Essential Role of Ters

Genetic studies show that the Rpps encoded by the PICIs interact with TerS of phages λ, ψ80, and the TerS-21, encoded by CTF073 prophage 4 [29]. Rpp binds TerS monomers to form heterodimers. The Rpp–TerS association blocks TerS from binding phage R sites in *cosB* and enables binding of *cos*-PICI. The TerS component of the Rpp–TerS dimer is strictly required, as a helper phage lacking functional TerS is unable to package PICI DNA [2]. Sequence analyses and genetic studies show that PICI-A contain a functional *cos* analogous to the *cos*es of λ-like helper phages (Figure 2B).

### 2.1. The PICI cos

Alignment of the DNA sequences of phage λ DNA and four PICI elements identified by Fillol-Salom et al. [1,2] is shown in Figure 2B. Remarkably, these PICI *cos*es have high identity in the *cosQ cosN,* and the spacing between these elements closely matches those in the helper phage *cos*es.

Since the distance between R3 and *cosN* in λ is critical for proper nicking and packaging [25,26], it is appropriate to assume that the same will hold for the PICIs. Therefore, the interval between a putative R3 equivalent in a PICI and the *cosN* will be close to that in phages λ and N15, for which there is information about the extent of R3. We propose that the sequences encompassing bp 48 to 65 of the PICIs, which are in equivalent positions to R3-λ and R3-N15, contains the Rpp-binding site, R3-PICI. The Rpps in the Rpp–TerS complexes are proposed to bind specifically to R3-PICI sites. Given the absence of I1, R2, and R1 sequences, PICIs *cosB*s architecture resembles *cosB*-N15, which contains only an R3 equivalent [19,20]. Interestingly, the putative R3 sequences of *cos*-PICI-A and *cos*-PICI-C differ in just a few positions. These differences are at bp 51 and 52. Other differences occur upstream of the R3 equivalent in PICI sequences, at positions 45 and 46 (Figure 4).

### 2.2. Rpp Proteins

The crystallographic structure of RppC shows a DBD, residues 1-64, that is remarkably like the DBDs of TerS-λ and the MetR transcription activator [30] and other members of the LysR group. The predicted DBDs of RppC and its relatives RppA, RppB, and RppG have two α-helices, α1 and α2, and two short β strands forming a “wing”, with a positively charged residue at the tip. These elements form a canonical winged α-helix turn α-helix DNA-binding motif. Thus, it has been proposed that Rpps’ α2-helixes bind to the major groove of an R sequence equivalent to R3 of phage λ [1]. Residues R21, T22, R25, K29, and R30 of the α2-helix would interact with bases in the major groove, with β hairpin residues K39 and K41 making minor groove contacts (Figure 4). However, these authors did not determine the precise region of the DNA in PICIs to which Rpps would bind.

### 2.3. Proposals for Rpp-R3 Binding: (1) Location of R3-PICI; (2) Rpp Wing Variation May Reflect Differences in DNA-Binding Specificity

The R3 segments of the four PICI coses display an identical sequence from positions 55 to 63 (Figure 4). This is a striking arrangement since, as described in the introduction, it has been demonstrated by NMR that TerS of phage λ contacts bases 55 to 63 [22]. Therefore, it is likely that all the Rpps (in the Rpp–TerS dimers) studied contact, primarily, bases at the same positions. It is tempting to propose that the differences in the 5′ side of the R3-equivalents of the various PICI *cos*es would provide specificity of binding by the corresponding Rpps since their residue differences are located mostly in the C-end of the α2-helix, in concordance with the orientation of the α1-helix of TerS-λ bound to λ-Rs [12,27,28]. In fact, the α2-helix of λ-TerS binds with its C-terminal end at the 5′ of R3 [31,32], and the same orientation has been proposed for the α2-helix of the TerS-Rpp complex [1].

Thus far, however, the only data available have established that RppA is essential for PICI-A packaging, and *cos*-binding specificity for the other Rpps is lacking. The conservation of R3-equivalents bp 55 to 63 in various PICIs, including islands from different bacterial species, suggests that they are all probably variations of a common ancestor, in the presence of some constraints. The variability in the Rpps includes residues that do not interact directly with the DNA [1]. Because, excepting RppA, evidence for the other Rpps promoting packaging of their respective PICIs is lacking, the variability could be attributed to genetic drift. In fact, the β-sheets of the “wing” of phages λ and 21 TerS proteins DBDs can be interchanged without affecting the phage burst sizes despite their numerous base pairs differences [31]. In addition, a K35-to-R35 substitution in λ TerS does not alter the phage phenotype [28].

### 2.4. Hypothesis: The Role of TerS in Hyjacking Is to Recruit TerL to cos-PICI

Since TerS is required for the hijacking process, what is TerL’s role in the process? This is puzzling and interesting because TerS is a specific DNA-binding protein whose role is to anchor TerL to the phage’s *cos*, ensuring accurate nicking of *cosN*. In hijacking, the Rpp–TerS heterodimer directs TerL away from *cos*-λ and to *cos*-PICI. Accordingly, the ability of TerS to bind *cos*-λ is not used. The TerS-λ DBD also binds DNA non-specifically, so one possible function for TerS might be to bind to PICI DNA non-specifically [9]. At this point, it is not clear whether *cos* is shaped into a hairpin by Rpp–TerS, nor is it clear whether the Rpp:R3-PICI interaction is sufficient for anchoring TerL for *cosN* nicking.

Unfortunately, no information is available on the structure of nicking/packaging initiation complex for phage N15, whose *cosB* consists solely of an R3 sequence. An alternative explanation for the role of TerS concerns the TerL-binding domain of TerS. As described above, this TerL-binding domain is proposed to be in the C-terminal 40 amino acids of TerS. The four characterized Rpp proteins may not have such a domain, as their C-termini are about 30–40 residues shorter than the λ-like TerS molecules (Figure 3). While it is possible that the Rpp proteins have developed an alternative way of binding to TerL, we find no amino acid sequences conserved among the Rpp proteins that might represent a conserved TerL-binding domain. Therefore, the simpler explanation is that the Rpps lack a TerL-binding domain and that the role of TerS is to provide TerL for packaging initiation (Figure 5). It is apparent that, in principle, Rpp_2_:TerL_1_ protomers, with their winged-HTH domain, have the full capacity to bind R3-PICI *cos*es on their own. However, there is no evidence that Rpps, as monomers or homodimers, do bind to PICI-R3 prior to TerS binding. On the other hand, two-hybrid analysis shows that Rpps form homodimers, as TerS does, but also TerS_2_:Rpp_2_ hetero-tetramers [1]. Whether this tetramer is able to bind PICI-R3 or assembles to form a protomer with a putative quaternary structure of Rpp_2_:TerS_2_:TerL_1_ prior to DNA binding at *cosN* and R3 is unknown. Irrespective of the pathway of assembly of a nicking/packaging initiation complex, the essential role of TerS in the heterodimer would be to bring together PICI-R3-bound Rpp and TerL and thus provide the proper DNA-binding specificity for *cosN* cutting (Figure 5). Thus, Rpps have two functions: to form heterodimers with TersS and to bind to an Rpp-specific R3, which is different from the helper phage.

## 3. Architecture of the *cosN*-PICI Nicking Complex

As a first approximation, it may be that the reason for the multiple separated sites in *cosB-*λ is to increase the DNA recognition specificity for the assembly of the initiation complex. Alternatively, we suspect that the structure is akin to an “energy ladder” in the sense that a large ΔG change is portioned into a series of smaller ΔG steps. Not only could this facilitate formation of the initiation complex, but it could more importantly enable the disassembly during the transition to a translocation complex after *cosN* nicking. In accordance with this reasoning, it has been found that λ TerS (gpNu1) does not bind R sites strongly and that a fragment of gpNu1 containing the DBD binds non-specific DNA and specific DNA about equally well [9].

*cosB* of λ-like phage N15 lacks R2, as is the case for the PICIs [1]. Therefore, in both cases, the ΔG ladder would be composed of only two steps: binding to *cosN* and R3. TerS-N15 DBD is dimeric [19]. Presumably, one of the TerS molecules binds to R3 specifically, but in the absence of I1 and R2, the other subunit of the TerS dimer may not bind DNA or could bind non-specifically perhaps to DNA in a region corresponding to R2 in phage λ DNA. It remains puzzling the necessity (or advantage) for the TerS-N15 dimer to bind, in addition to R3, to a non-specific DNA site. The ability of phage N15 to package λ DNA shows that N15 TerS recognizes λ R3 [19,20]. If dimeric TerS-N15 indeed binds non-specifically to a second site, then the presence of R2 in λ DNA would be irrelevant. Alternatively, it could bind specifically, increasing the efficiency of DNA capture. Packaging of λ R2 mutant DNA by N15-specific terminase could elucidate this question. We note that λ cannot package N15 DNA. Thus, TerS-λ requires the presence of both R3 and R2 to function, likely due to a lower R3 affinity than that of TerS-N15.

## 4. Implications for Phage DNA Packaging

The proposal that Rpp binds indirectly to TerL, using TerS as a bridge, poses a challenge to the classical model structure of the initial nicking complex. For example, the early model of Becker and Murialdo proposed that the TerS molecule bound to R3 anchored TerL for *cosN* cleavage [32]. This model was proposed before the discovery that a TerL_1_:TerS_2_ protomer was a fundamental unit of terminase [17,33]. Our proposal that TerL is anchored by a TerS not bound to R3 challenges this earlier picture and raises an architectural question about the phage nicking complex and about how TerL is bound in the TerL_1_:TerS_2_ protomer. Proposing that TerL is bound by TerS of the Rpp:TerS heterodimer implies that a single TerS is sufficient to position TerL for nicking at the proper places in *cosN*. The nature of the TerS–TerL interaction is unknown at the molecular level. The more complex set of interactions, R3:Rpp:TerS:TerL:TerL challenges the notion that the initiation complex is a rigid structure engineered to place TerL for accurate *cosN* nicking. An alternative view is that the complex at *cos* has a function of localizing TerL to *cos*, preventing TerL from introducing harmful nicks in the bacterial chromosome. The positioning of TerL for precise nicking of *cosN* might be due to local interactions between *cosN* and TerL.

## Figures and Tables

**Figure 1 viruses-14-00818-f001:**
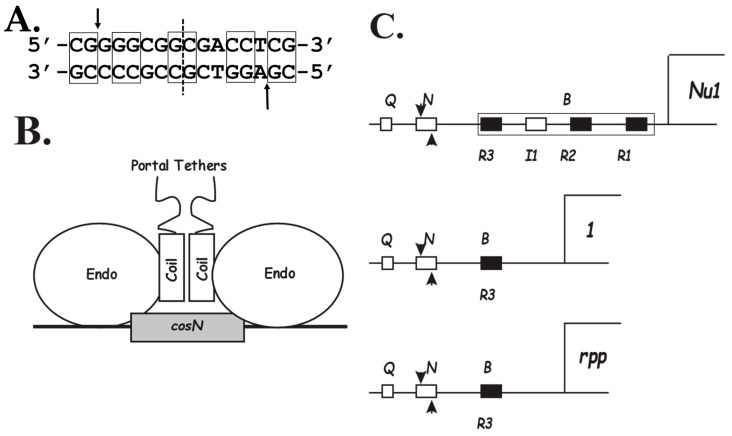
*cos* organization in λ-like phages and PICIs. (**A**). Sequence of *cosN*-λ: boxed bp show twofold rotational symmetry, where the dashed vertical line indicates the center of symmetry. Arrows indicate nicking sites. Note two rotationally symmetric flanking bp on both sides of the cohesive end sequence. (**B**). Cartoon of the proposed arrangement of the TerL endonuclease centers on *cosN*. Distal to the endonuclease centers are α-helixes proposed to form a parallel coiled coil, followed by C-terminal tethers for docking on the portal [7,8]. (**C**). *cos* organizations of λ, N15, and a PICI. Arrowheads indicate nick positions. Top: *cosB*-λ includes IHF site I1 and three R sequences. The start codon for TerS-encoding gene *Nu1* is at bp 191. Middle: *cos* of phage N15 contains a single R sequence, R3, and no IHF site. The start codon for TerS-encoding gene *1* is at bp 89. Bottom: Arrangement of *cos* and the *rpp* gene in phage-induced pathogenicity islands (PICIs). The *rpp* start codon for PICI-C is at bp 102.

**Figure 2 viruses-14-00818-f002:**
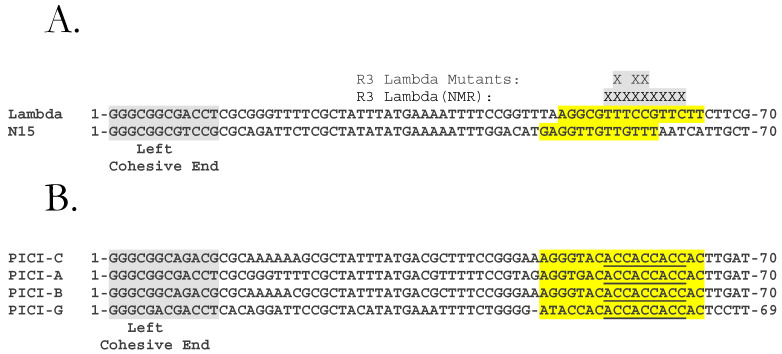
*cos* Alignments and R3 sites. (**A**) Alignment of the left ends of λ and N15 DNAs. Sequences start at bp 1, the first base of the left cohesive end. The 12 bp left cohesive end sequence is grey-highlighted. The position of R3-λ, bp 50–65, based on sequence identity with R2 and R1, is highlighted in yellow. Additionally shown is the protein-interactive segment bp 55–63, as determined by NMR. Three R3-λ bp, 56, 58, and 59, have functional roles in DNA packaging as defined by genetic and biochemical studies. The location of R3-N15, defined by linker scanning mutagenesis, bp 48–60, is highlighted in yellow. (**B**). The proposed R3 segments, bp 48–65 for PICI-A, PICI-B, PICI-C, and bp 47–64 for PICI-G, are highlighted in yellow. The left cohesive ends are highlighted in grey, and the bp 55–63 segments of the *cos*-PICIs are underlined.

**Figure 3 viruses-14-00818-f003:**
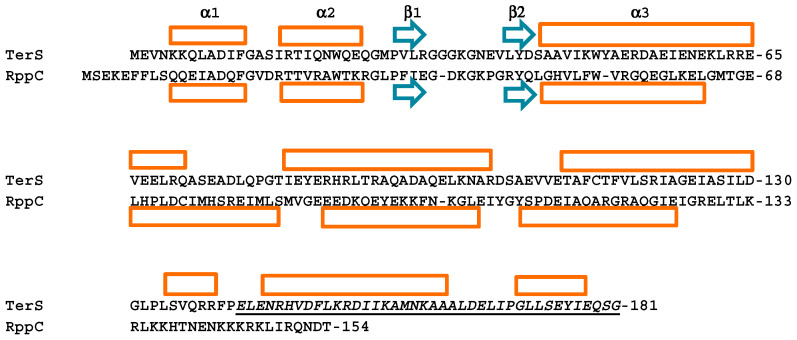
Secondary structures of TerS-λ and RppC. The top alignment shows the DBDs of the two proteins. The TerS-λ DBD structure is from DeBeer et al. [22] and Fillol-Salom et al. [1] and the RppC structure if from Fillol-Salem [1]. The TerS-λ structure or residues 66 to 181 is the JPred prediction (www.compbio.dundee.ac.uk/jpred/, accessed on 1 December 2021). α-helixes are indicated by orange rectangles and β-strands by blue arrows. TerS-λ residues proposed to form the TerL-binding domain are underlined and italicized. RppA, B, C, and G have 135, 146, 153, and 139 residues, respectively.

**Figure 4 viruses-14-00818-f004:**
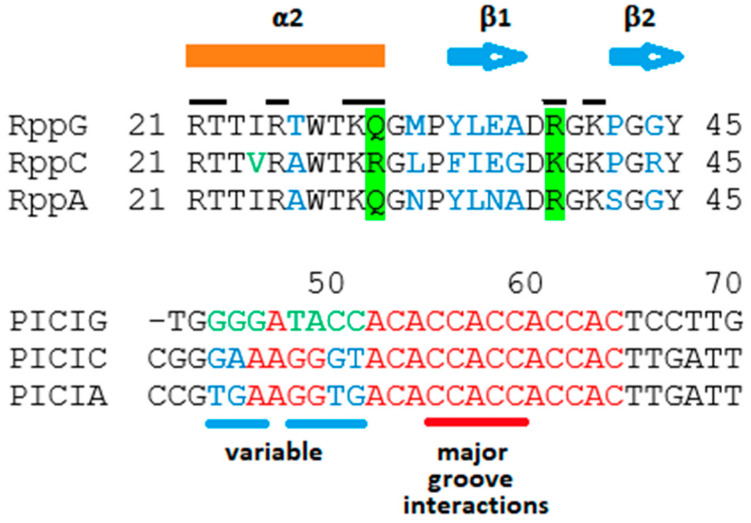
Possible recognition specificity of PICI’s R3 and their correspondent Rpps. (Top) Alignment of studied Rpps (RppB has an identical residue sequence to RppA in the shown region). The bar on top shows the extent of the α2-helix and the arrows those of the β-sheets. The lines on top of the one-letter code show the residues that have been proposed to interact with bases in R3. Of those residues, the ones highlighted in green differ between RppC and the other Rpps. Therefore, they may provide R3-binding specificity. The residues in blue show differences in other regions, and some of them may provide specificity for RppG and RppA to discriminate from PICIG- and PICIA-R3s. Notice that most of the divergence is in the α2-helix–β1-sheet link, especially in the β1-sheet itself and in two residues of the β2-sheet. (Bottom) Alignment of the correspondent R3 regions of *cosB* PICIs. Red bases show identity, blue ones show differences between A- and C-R3s, and green bases show difference between PICIG-R3 and the other two Rs. Notice the striking identity in the 3′ part and the deviation in the 5′ portion. The sequence that varies between the PICIs R3 are underlined in blue. The base pairs that are supposed to interact with Rpps residues in the major groove are underlined in red.

**Figure 5 viruses-14-00818-f005:**
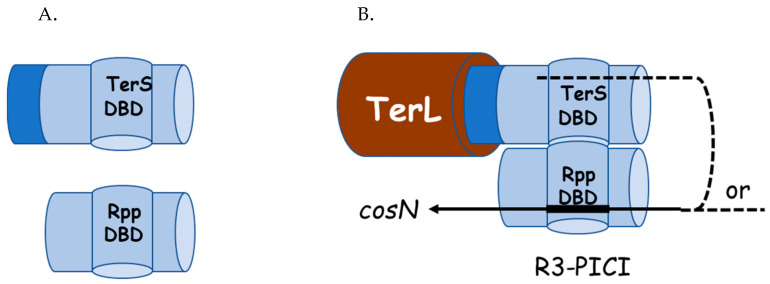
Proposed architecture of the Rpp-Ter sponsored initiation complex at *cosB*-PICI. (**A**). Domain structures for monomeric TerS and Rpp. The Rpp and TerS DBDs and oligomerization domains are small and large blue cylinders, resp. Dark blue represents the TerL-binding domain of TerS. (**B**). Proposed arrangement of the Rpp–TerS–TerL complex at *cos*-PICI. Components colored as in A with brown cylinder represent the N terminus of TerL.Solid line represents PICI DNA. Dashed lines indicate uncertainty about whether PICI DNA contacts the DBD of TerS. The proposed architecture of TerL at *cosN* is shown in Figure 1B.

## Data Availability

Not applicable.

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
