# Peer review of "Enteric Chromosomal Islands: DNA Packaging Specificity and Role of λ-like Helper Phage Terminase"

_viruses, 2022, doi:10.3390/v14040818_

Round 1

Reviewer 1 Report

I really enjoyed reading this manuscript. In this work, the authors revisited the work that the Rpp proteins do promoting the transfer of the E. coli PICIs. Addiitonally, they also proposed interesting hypothesis about how these proteins perform their function, which I'm convinced will be really relevant for the people working on this area of research. Overall, this is a very nice paper.

Author Response

We thank the reviewer for the careful reading of our manuscript.  As there are no comments requiring a response, we simply thank this reviewer.  

Reviewer 2 Report

This was an extremely dense read (actually several reads were necessary). Most individual sentences are perfectly clear but the overall direction that the ms is heading is completely lost among the very detailed information that the authors present. I have no concerns about the conclusions ultimately made, they seem perfectly reasonable, but I would not be surprised if many readers quickly lose interest and give up before coming to the conclusions or even understanding what hypothesis was being proposed.

The title should explicitly describe the hypothesis, and the abstract should expand but focus on that idea. Much of the central portion of the abstract is better suited for the Intro or elsewhere in the paper.

Although not relevant for the details of the hijacking mechanism under consideration, casual readers may benefit from a couple of lines in the Intro on other hijacking scenarios, such as the P2/P4 paradigm and the SAPIs. It may help readers think about the bigger picture of how many parasites are themselves subject to parasitism.

Most subsections should start with a statement describing how the details that follow relate to the hypothesis being proposed.

Fig. 2 and its legend don’t agree. Perhaps during pdf conversion, sequence annotations and/or highlighting seem to have slipped or are missing

Alternatively, the manuscript could be positioned as a review of cos recognition by Ter, and the sequestering of TerS by RppA, and conclude with the unifying hypothesis.
